# Graph Sampling-Based Multi-Stream Enhancement Network for Visible-Infrared Person Re-Identification

**DOI:** 10.3390/s23187948

**Published:** 2023-09-18

**Authors:** Jinhua Jiang, Junjie Xiao, Renlin Wang, Tiansong Li, Wenfeng Zhang, Ruisheng Ran, Sen Xiang

**Affiliations:** 1College of Computer and Information Science, Chongqing Normal University, Chongqing 401331, China; 2021210516040@stu.cqnu.edu.cn (J.J.); 2022210516103@stu.cqnu.edu.cn (J.X.); tiansongli@cqnu.edu.cn (T.L.); 2School of Computer Engineering, Weifang University, Weifang 261061, China; wfuwrl@126.com; 3School of Information Science and Engineering, Wuhan University of Science and Technology, Wuhan 430081, China; xiangsen@wust.edu.cn

**Keywords:** Multi-Modal Data, VI Re-ID, modality discrepancy, Contour Expansion Module, Cross-modality Graph Sampler

## Abstract

With the increasing demand for person re-identification (Re-ID) tasks, the need for all-day retrieval has become an inevitable trend. Nevertheless, single-modal Re-ID is no longer sufficient to meet this requirement, making Multi-Modal Data crucial in Re-ID. Consequently, a Visible-Infrared Person Re-Identification (VI Re-ID) task is proposed, which aims to match pairs of person images from the visible and infrared modalities. The significant modality discrepancy between the modalities poses a major challenge. Existing VI Re-ID methods focus on cross-modal feature learning and modal transformation to alleviate the discrepancy but overlook the impact of person contour information. Contours exhibit modality invariance, which is vital for learning effective identity representations and cross-modal matching. In addition, due to the low intra-modal diversity in the visible modality, it is difficult to distinguish the boundaries between some hard samples. To address these issues, we propose the Graph Sampling-based Multi-stream Enhancement Network (GSMEN). Firstly, the Contour Expansion Module (CEM) incorporates the contour information of a person into the original samples, further reducing the modality discrepancy and leading to improved matching stability between image pairs of different modalities. Additionally, to better distinguish cross-modal hard sample pairs during the training process, an innovative Cross-modality Graph Sampler (CGS) is designed for sample selection before training. The CGS calculates the feature distance between samples from different modalities and groups similar samples into the same batch during the training process, effectively exploring the boundary relationships between hard classes in the cross-modal setting. Some experiments conducted on the SYSU-MM01 and RegDB datasets demonstrate the superiority of our proposed method. Specifically, in the VIS→IR task, the experimental results on the RegDB dataset achieve 93.69% for Rank-1 and 92.56% for mAP.

## 1. Introduction

Person re-identification (Re-ID) [1,2,3,4,5,6,7] is a complex computer vision task that focuses on matching individuals across non-overlapping camera views. The main objective is to associate images or videos of the same person while maintaining a low recall rate, thereby reducing the likelihood of incorrect matches. Effective Re-ID techniques have significant applications in various domains, such as surveillance, security, and public safety. With the increasing demand for Re-ID, there is a need to match infrared person images captured under challenging lighting conditions with visible person images. Consequently, VI Re-ID [8,9,10,11,12,13] garners significant attention from both the industry and academia.

Besides the intra-modal variations already present in single-modal Re-ID, a key challenge in the VI Re-ID task is how to reduce the modality discrepancy between visible and infrared images of the same identity. Existing research approaches primarily rely on modal transformation methods. These methods generate cross-modal or intermediate-modal images corresponding to person images to convert heterogeneous modalities into a unified modality, thereby reducing the modal discrepancy. Specifically, Generative Adversarial Networks (GANs) [14] and encoder–decoder structures [15,16] are commonly introduced in these methods. However, the transformation from infrared images to visible images is ill-posed, which may introduce additional noise and fail to generate accurate visible images. Moreover, GAN-based models [17] often overlook the relationships between global features or local features of person images in the VI Re-ID task, leading to limited modal adaptability in their methods.

To enhance the method’s adaptability to cross-modal challenges, some recent researchers apply modal-shared feature learning to the VI Re-ID task, which projects visible and infrared images into a specific shared embedding space, achieving cross-modal feature alignment. These approaches can be further divided into global feature learning and local feature learning. Specifically, global feature learning represents a person image as a single feature vector, which is suitable for capturing overall person identity information. On the other hand, local feature learning uses a set of feature vectors based on parts or regions to represent the image, allowing for a more detailed capture of the local features of person images. In addition, the two-stream convolutional neural network architecture is commonly applied in such methods and combined with loss functions (such as identity loss, triplet loss, etc.) for constraint.

Although these methods have achieved good results in alleviating the modality discrepancy, they still have certain limitations: (1) Existing modal-shared feature learning methods typically focus on exploring either global or local feature representations and rarely combine the advantages of both features. Moreover, due to the nature of infrared images, which only contain a single channel reflecting objects’ thermal radiation, key features like color cannot be utilized for cross-modal matching. Directly extracting features from infrared images may also suffer from interference from identity-irrelevant information. (2) These methods all adopt some basic sampling techniques, such as random sampling and uniform sampling, which do not consider the relationships and similarities between samples with different modalities. Additionally, in VI Re-ID tasks, there are limited features that can be extracted when dealing with infrared modality retrieval, resulting in numerous similar features among samples from different classes. Consequently, conventional sampling methods struggle to capture the subtle differences between these similar features.

To address the aforementioned two issues, we propose a novel method named the Graph Sampling-based Multi-stream Enhancement Network (GSMEN) in this paper. It is noteworthy that when humans engage in the visual judgment of infrared surveillance, they heavily rely on contour information. Despite the absence of color and texture features in infrared images, contour and shape information remains clear and visible, as depicted in Figure 1a with the contour image. It is evident that contours exhibit certain cross-modal invariance between visible and infrared images [18]. Additionally, as contours provide a holistic representation of a person rather than localized features, their global features can better capture the characteristic information. This observation motivates us to extend the global features of contours to the local features obtained from modal-shared feature learning, with the aim of enhancing the feature representation capability and reducing the cross-modal discrepancy between visible and infrared modalities. Consequently, the Contour Expansion Module (CEM) is proposed to fuse the contour-enhanced features with local features, resulting in improved matching performance for cross-modal image pairs.

Then, to tackle the challenge of exploring the boundaries between hard classes in VI Re-ID task, an efficient batch sampling technique is introduced, known as the Cross-modality Graph Sampler (CGS). Specifically, CGS involves constructing nearest neighbor relationship graphs for all classes in the visible and infrared modalities at the beginning of each epoch and then combining them. Subsequently, CGS conducts batch sampling by randomly selecting a sample as the anchor and choosing its top-*k* nearest neighboring samples of different classes, each class containing the same number of S instances, as illustrated in Figure 1b. Therefore, CGS ensures that the samples within a batch are mostly similar to each other, providing informative and challenging examples for discriminative learning. This sampler aims to explore the boundary relationships between hard classes and enhance the discriminative power of the learned model.

In summary, our contributions in the paper are as follows:To enhance feature representation and reduce the cross-modal discrepancy between visible and infrared modalities, we propose the Contour Expansion Module (CEM), which combines the global features of contours with the local features obtained from modal-shared feature learning. To the best of our knowledge, this method represents the first attempt at tackling the VI Re-ID task.To explore the boundaries between hard classes, we introduce the Cross-modality Graph Sampler (CGS). The sampler constructs nearest neighbor relationship graphs separately for the visible and infrared modalities, and then combines them for batch sampling. This sampling strategy ensures that samples within a batch are mostly similar to each other, providing informative and challenging examples for discriminative learning.We conduct experiments on large-scale VI Re-ID datasets, SYSU-MM01, and RegDB. The results demonstrate that our method achieves significant improvements in matching performance and modal adaptability.

## 2. Related Work

VI Re-ID aims to address not only the challenges of handling intra-modal differences but also the cross-modal disparities arising from heterogeneous images. Therefore, alleviating cross-modal disparities is crucial, as they can exacerbate existing intra-modal differences.

To tackle these challenges, researchers attempt the modality-shared feature learning approach [19,20,21,22,23], which focuses on extracting discriminative and robust features from heterogeneous modalities for the model’s learning process. For instance, Wu et al. [8] introduce the large and challenging benchmark dataset (SYSU-MM01) and propose a deep one-stream zero-padding network for RGB-IR image matching. Additionally, Fu et al. [19] present a cross-modality neural architecture search method to enhance the effectiveness of neural network structures for VI Re-ID tasks. Furthermore, Zheng et al. [24] adopt eight attributes as annotation information in the PAENet to learn detailed semantic attribute information.

In recent years, image generation-based methods [14,25] are also applied to the VI Re-ID task, aiming to narrow the gap between visible and infrared modalities by adopting modality auxiliary information. For this purpose, Dai et al. [13] introduce a framework based on Generative Adversarial Networks (GANs) [14] for cross-modal image generation and propose cmGAN for feature learning. Similarly, Wei et al. [26] propose a comprehensive modality generation module that combines features from different modalities to create a new modality, effectively integrating multi-modal information. Additionally, Lu et al. [25] introduce the Progressive Modality-shared Transformer (PMT), which employs grayscale images as auxiliary modalities to enhance the reliability and commonality of visual features across different modalities, addressing the negative effects of modality disparities.

Furthermore, it is worth noting that among various types of information in cross-modal images (such as color, texture, and contour), contour information is crucial in cross-modal retrieval, as it exhibits strong modality invariance. However, previous researchers did not consider using it as auxiliary information to alleviate modality differences. Therefore, we attempt to introduce it as auxiliary enhancement information to improve the image-matching capability in VI Re-ID tasks.

## 3. Method

Our VI Re-ID framework is based on a two-stream convolutional neural network [4] but incorporates contour information to enhance the model’s cross-modal adaptability. The method we propose is outlined as shown in Figure 2, and the specific steps are as follows: First, the Cross-modality Graph Sampler (Section 3.1) samples the dataset with the aim of including categories that are close in distance into the same batch. The obtained samples consist of two modalities: visible and infrared modalities, both of which are obtained through contour extraction to produce their respective contour images. Next, these four categories (two types of original images and their corresponding contour images) of images are separately input into their corresponding backbone networks. Finally, the resulting sample features and contour features are fused in the Contour Expansion Module (Section 3.2) to mitigate the differences between the visible and infrared modalities.

### 3.1. Contour Expansion Module

The goal of contour detection is to identify pixels in the image that correspond to regions with significant changes in grayscale values. Recently, there have been studies [23,25] applying contour detection to object detection and semantic segmentation, achieving success. The challenge in the VI Re-ID task lies in the significant differences between the visible and infrared modalities. Additionally, contour information exhibits strong modality invariance. Hence, this inspired us to apply contour detection to the VI Re-ID task to alleviate the modal discrepancies. First, the pre-trained SCHP (Self-Correction Human Parsing) [29] is adopted as the contour detector to segment person contour maps from the images. Taking the visible image for example, the contour detection of visible image $X^{vis}$ is below:(1)$$X^{visc}=\sigma (X^{vis}),$$
where σ(⋅) denotes the contour detector, $X^{visc}$ represents the person contour map generated from the visible image $X^{vis}$. Then, the obtained contour information needs to be integrated into the original image information. However, the fusion methods [30,31,32] vary in diversity. Therefore, this paper investigates the impact of different fusion methods on the experiments, including Element-wise addition and concatenation, as shown in Figure 3. Specifically, Element-wise addition emphasizes the employment of contour feature to supplement person image-related semantic information, while Element-wise concatenation expands the feature dimension without losing the respective information of person image and contour. Here are the specific fusion methods employed in this study.

Element-wise addition. As shown in Figure 3b,c, we, respectively perform feature addition before the CNN and after Conv Block *n* (1≤n≤5). The specific formula is as follows:(2)$$F_{i}^{vism}=F_{i}^{vis}+F_{i}^{visc},$$
where $F_{i}^{vism}$ represent visible feature after merging contour feature $F_{i}^{visc}$ to basic visible feature $F_{i}^{vis}$. i∈{RGB, conv-1, …, conv-5} represents the fusion methods shown in Figure 3b,c. Specifically, $F_{RGB}^{vism}$ represents the three-channel features obtained after transforming the RGB image into RGB data.

Element-wise concatenation. As depicted in Figure 3a, we augment the local features by incorporating the global feature of the visible image contours. Firstly, the visible image and the contour image are separately processed through CNN to obtain features Fconv-5vis∈ℝc×h×w and Fconv-5visc∈ℝc×h×w, respectively. Next, the output features are subjected to Generalized Mean Pooling for local and global feature pooling, resulting in features Flocalvis∈ℝc×h/p and Fgobalvisc∈ℝc:(3)Flocalvis=GEMPooling(Fconv-5vis,(h/p,w)),
(4)Fgobalvis=GEMPooling(Fconv-5visc,(h,w)),
where GEMPooling(z,(x,y)) applies Generalized Mean Pooling [33] to *z* using a two-dimensional scale with the height of *x* and width of *y*. *p* is the number of local body parts in visible image. Then, 1 × 1 convolutional layers are utilized to adjust the number of feature channels in Flocalvis and Fgobalvisc to *C*. Finally, by concatenating the local feature Fvis of the visible image with its contour image’s global feature Fvisc, the new visible feature Fvism is obtained:(5)Fvism=concat(Fvis,Fvisc),
where concat(e,f) represents the concatenation of feature *e* and feature *f*. Considering the comparative experiments in Section 4.5.2, Element-wise concatenation better accomplishes contour enhancement than other ways and is chosen as the fusion method for contour information in CEM.

### 3.2. Cross-Modality Graph Sampler

Both DG Re-ID [34,35,36,37,38] and VI Re-ID [5,9,10,11,12] face the challenge of modality differences. Furthermore, conventional sampling methods exhibit significant randomness, making it insufficient to distinguish the boundaries between hard classes. In contrast, the CGS sampler effectively addresses this limitation by focusing on grouping similar samples into the same batch. Combining with Figure 4, the details of CGS should be introduced below.

Before each epoch, we calculate the distances or similarities between classes using the latest trained model and then construct a graph encompassing all classes. This approach allows us to leverage the relationships between classes for informative sampling. To illustrate, one image per class is randomly chosen to form a smaller sub-dataset. Next, the feature Fvism∈ℝC×d should be extracted thought the latest trained model, where C represents the total number of training classes and d is the feature dimension. Subsequently, the pairwise Euclidean distances between all the selected samples are computed by the feature Fvism∈ℝC×d. As a result, we obtain a distance matrix distv∈ℝC×C that encompasses all classes:(6)distv=ϕ((Fvism,Fvism),dim=0),
where ϕ((x,y),dim=z) represents the pairwise Euclidean distances calculated for the feature vectors x and y after aligning them along the *z*th dimension. Similarly, the process is applied to the infrared modality sample set:(7)disti=ϕ((Finfm,Finfm),dim=0),
where Finfm∈ℝC×d is the feature extracted from Infrared samples. Afterwards, to obtain the neighboring classes across different modalities, the overall class distance matrix dist can be obtained by adding matrices distv and disti together:(8)dist=distv+disti.

Later, the top k−1 nearest neighboring classes need to be denoted by $N(c)=\{x_{i}|i=1,\;\ldots ,\;k-1\}$ from each class c, where k is the number of classes to sample in each mini-batch. Subsequently, a graph $G=\left(V,E\right)$ with $V=\left\{c|c=1,\;\ldots ,\;C\right\}$ represents the vertices, where each class corresponds to one node, and $E\;=\{\left(c_{1},\;c_{2}\right)\;|\;c_{2}\in N\left(c_{1}\right)\}$ representing the edges. Finally, according to the graph G, we perform random sampling of *S* instances per class to create a mini-batch containing B=K×S samples for training. This approach allows us to establish connections between classes based on their proximity, enabling informative sampling for our training process.

### 3.3. Loss Function

Triplet loss $L_{tri}$ [39] and Cross-entropy loss $L_{id}$ are the fundamental losses for image classification tasks. Moreover, the Barlow Twins loss $L_{ssd}$ [40] is a self-supervised learning loss function, which is also introduced into our method to improve its performance. The overall loss is computed as follow:(9)$$L=L_{id}+L_{tri}+L_{ssd}$$

## 4. Experiments

### 4.1. Datasets and Evaluation Protocol

#### 4.1.1. Datasets

SYSU-MM01 [8] dataset consists of 491 different identities captured by four visible cameras and two infrared cameras. It encompasses two search modes: All-Search mode and Indoor-Search mode. Specifically, in the All-Search mode, the gallery set comprises all images captured by the visible cameras, allowing researchers to explore scenarios where all available visible cameras are employed for Re-ID tasks. In contrast, the Indoor-Search mode utilizes images from indoor visible cameras as the gallery set, which is employed for studying Re-ID tasks in indoor environments. The training set comprises 19,659 visible (VIS) images and 1792 infrared (NIR) images, providing a diverse collection of data covering 395 distinct person identities. The test set consists of 3803 infrared images from 96 different person identities, serving as the query set.

RegDB [41] dataset comprises pairs of images captured by visible and infrared cameras. It contains images of 412 different identities, with each identity having 10 visible images and 10 infrared images. These images are captured by a pair of cameras that overlap with each other, providing a comprehensive set of data for evaluation. Additionally, in order to effectively validate various methods, the dataset offers two testing protocols: Infrared-to-Visible (IR-to-VIS) and Visible-to-Infrared (VIS-to-IR).

#### 4.1.2. Evaluation Protocol

To assess the performance of both datasets, we employ standard evaluation protocols, which incorporate Cumulative Matching Characteristics (CMC) [42], and mean Average Precision (mAP) [43] as evaluation metrics. Specifically, we conducted ten tests and computed the average results across these tests.

### 4.2. Implementation

The proposed methodology utilizes the PyTorch deep learning framework and is implemented on an NVIDIA RTX 3090 GPU. Building upon existing VI Re-ID methods, a pretrained ResNet-50 [28] is employed as a backbone network. During training, all images should be resized to the dimensions of 288 × 144 and data augmentation techniques (random cropping and random horizontal flipping) [44] are introduced.

The training process involves using the stochastic gradient descent (SGD) optimizer with a momentum value of 0.9. The initial learning rate is set to 0.01, and a warm-up strategy is employed to adjust the learning rate. Specifically, the learning rate is initialized to 0.01 and undergoes 10 decays, each occurring every twenty epochs. The training is stopped after 60 epochs. The number *p* of local body parts in Formula (3) is set to 6.

### 4.3. Comparison with State-of-the-Art Methods

In this section, a comparison between the proposed method and state-of-the-art VI Re-ID approaches is conducted, including Cross-modal feature learning: DDAG [21], AGW [4], cm-SSFT [45], GLMC [22] (best method), MPANet [12], LBA [46], CM-NAS [19], MMN [47], MID [48]; Modal transformation: cmGAN [13], AlignGAN [49], Xmodal [17], SFANet [50], AGMNet [51] (best method), and PMT [25]. According to Table 1, the results on the two datasets demonstrate that the proposed GSMEN outperforms state-of-the-art methods, achieving the outstanding performance. The specific comparisons are below:

Evaluations on SYSU-MM01. In comparison with the state-of-the-art Cross-modal feature learning method GLMC [22], our approach demonstrates significant superiority in both the All-Search and Indoor-Search modes. Specifically, in the All-Search mode, our method achieves a higher mAP and Rank-1 accuracy by 7.11% and 8.6%, respectively, compared to GLMC. Similarly, in the Indoor-Search mode, our method outperforms GLMC with a mAP and Rank-1 accuracy higher by 6.11% and 11.37%, respectively.

Furthermore, when compared to the current best Cross-modal feature learning method AGMNet [51], our approach also achieves remarkable performance gains. In the All-Search mode, our method surpasses AGMNet with a higher mAP and Rank-1 accuracy by 4.43% and 3.34%, respectively. Similarly, in the Indoor-Search mode, our method outperforms AGMNet with a mAP and Rank-1 accuracy higher by 1.83% and 3.54%, respectively.

Evaluations on RegDB. In comparison with the best-performing method in the VIS-to-IR mode, our approach demonstrates remarkable superiority. Specifically, our method achieves a higher mAP by 4.86% in this mode, showcasing its effectiveness in handling the cross-modal matching between visible and infrared images.

Similarly, in the IR-to-VIS mode, our method outperforms the best-performing method with a significantly higher mAP by 10.67%. This result highlights the exceptional capability of our approach to effectively address the challenges of cross-modal matching between infrared and visible images.

These impressive performance demonstrates the versatility and effectiveness of our proposed method in handling VI Re-ID task.

### 4.4. Ablation Study

In this section, a comprehensive ablation study is conducted to thoroughly evaluate the contributions of the Contour-oriented Enhancement Module (CEM) and the Cross-modality Graph Sampler (CGS) in our proposed approach. By systematically adding or removing these modules, we investigate their individual impacts on the performance of our model. The results are presented in Table 2, where shows the mAP and Rank-1 accuracy for each experimental setting.

First, we establish a baseline model [4] that comprises basic feature extraction and re-ranking with K-reciprocal [52]. In the subsequent analysis, the All-Search evaluation protocol on the SYSU-MM01 dataset [8] is applied as the benchmark for comparison. The performance of the baseline model achieves a mAP of 70.54% and a Rank-1 accuracy of 72.97%.

In the next step, to assess the influence of the CEM module, we integrate it into the baseline model. The inclusion of the CEM module results in significant performance gains, with mAP and Rank-1 accuracy increasing by 7.74% and 8.51%, respectively. This demonstrates that the CEM module effectively enhances feature representation and reduces modality discrepancy, contributing to the overall improvement in VI Re-ID performance.

Then, the impact of the CGS module is evaluated by incorporating it into the baseline model. The addition of the CGS module also leads to notable performance improvements, with mAP and Rank-1 accuracy increasing by 2.45% and 1.61%, respectively. The CGS module facilitates informative and challenging sample selection, effectively optimizing the training data and further enhancing the model’s discriminative capability.

Finally, we examine the combined effect of both the CEM and CGS modules by integrating them into the baseline model simultaneously. Remarkably, this joint integration yields remarkable performance enhancements, with mAP and Rank-1 accuracy increasing by 10.24% and 9.73%, respectively. The synergistic interplay between CEM and CGS reinforces the feature representation and sample selection aspects, leading to substantial overall improvements in the VI Re-ID task.

The ablation study demonstrates the effectiveness and significance of both the CEM and CGS modules in our proposed approach. The CEM module successfully leverages contour information to enhance feature representation, while the CGS module optimizes the sampling strategy for informative and challenging examples. By understanding the individual contributions of these modules, our study offers valuable insights into the design of a robust and efficient VI Re-ID task.

### 4.5. Comparison Experiment

#### 4.5.1. Comparison Experiment of Sampling Methods

In our comparative experiments on different sampling methods, namely Random Sampler, Uniform Sampler, and our proposed Cross-modality Graph Sampler (CGS), we observe significant differences in their performances from Figure 5. Specifically, our CGS sampling method outperformed both Random Sampling and Uniform Sampling by a notable margin. The mAP achieved with CGS is 1.81% higher than Random Sampling and 2.41% higher than Uniform Sampling on the SYSU-MM01 dataset with All-Search mode. This result clearly demonstrates the superiority of our CGS sampling approach in improving the overall performance of the VI Re-ID task. Compared to other sampling methods that exhibit randomness, CGS leverages the relationships among classes and ensures that instances within a batch are mostly similar, providing informative and challenging examples for discriminative learning. By incorporating such informative sampling, our method is better able to handle cross-modal challenges and effectively capture subtle differences between similar features, leading to the improved performance observed in the experiments (Table 3).

#### 4.5.2. Comparison Experiment of Fusion Methods

Besides, considering Section 3.1, for the method of fusing contour information, comparative experiments need to be conducted on different feature fusion methods, such as Element-wise addition and Element-wise concatenation (EC). Therefore, some comparative experiments are shown in Figure 5a, which can help determine which fusion method performs best in the task. Specifically, when our model adopts the Element-wise Concatenation fusion method, the mAP and Rank-1 accuracy of 72.97% and 70.54% is achieved, which outperformed other fusion methods and showed the best overall fusion performance. Taking into account the above analysis, the Element-wise Concatenation fusion method is employed for integrating contour information in this paper.

#### 4.5.3. Comparison Experiment of the Contour Detectors

For the selection of contour extraction methods, two options are considered: Canny edge detection and Self-Correction Human Parsing (SCHP). Therefore, a comparative analysis is conducted, and the specific results are shown in Table 4. Specifically, under the Indoor-Search mode, the SCHP method outperformed the Canny edge detection method with a Rank-1 accuracy improvement of 7.93% and an mAP improvement of 5.65%. These results indicate that the SCHP method is better suited for contour extraction in our approach.

#### 4.5.4. Comparison Experiment of the Output Channel

Furthermore, before employing the Element-wise Concatenation fusion method to merge the contour global features with the local features obtained from modal-shared feature learning, the output channel values of the contour global features are also worth our attention. In Figure 5b, we compare the output channel values of contour global features from 1 × 1 Conv. Our model achieves better performance when the output channel is set to 512. This result indicates that setting the output channel to 512 enhances the representation capability of our model for fusing contour information. This finding confirms the significance of selecting an appropriate output channel value for the effective utilization of contour global features and the enhancement of model performance.

### 4.6. Qualitative Analysis

In this section, we compare our proposed method with the AGW [4] approach using the SYSU-MM01 dataset. For the comparison, two sample images are selected, one depicting the frontal view and the other showing the rear view of individuals, as query samples. The Rank-10 visualization results are presented in Figure 6.

Upon analyzing the results, we observe that our method, with the inclusion of contour information and the utilization of the CGS sampler, effectively improves the retrieval performance. Specifically, in the case of the AGW method, there are instances of two erroneous matches in the rearview image retrieval, where these two images belong to the same class. In contrast, our method achieves correct Rank-10 results for all samples, indicating the superiority of the CGS sampler in distinguishing between similar samples from different classes. Moreover, when using the frontal view as a query sample, the AGW method shows three incorrect matches. These errors can be attributed to the similarity in backgrounds between these samples and the query sample. However, there are noticeable differences in the contour information between them. After enhancing the contour information in our method, only one matching error was observed, showcasing the significant role of contour assistance in enhancing matching capability.

Overall, these qualitative analyses demonstrate that the integration of contour information and the utilization of the CGS sampler effectively address the challenges posed by modal discrepancies and improve the precision and accuracy of the VI Re-ID task.

## 5. Conclusions

In this paper, we propose the Graph Sampling-based Multi-stream Enhancement Network (GSMEN) for the VI Re-ID task. The GSMEN integrates contour information with the globally shared contour features obtained from modal-shared feature learning. This integration aims to enhance feature representation and reduce cross-modal discrepancy. Our approach introduces the Contour Expansion Module (CEM) for fusing contour-enhanced features with local features and the Cross-modality Graph Sampler (CGS) for effective batch sampling. Experimental results on large-scale datasets demonstrate significant improvements in matching performance and modal adaptability. Our contributions include the novel CEM approach and the efficient CGS sampler, which show promising potential for VI Re-ID in various applications.

## Figures and Tables

**Figure 1 sensors-23-07948-f001:**
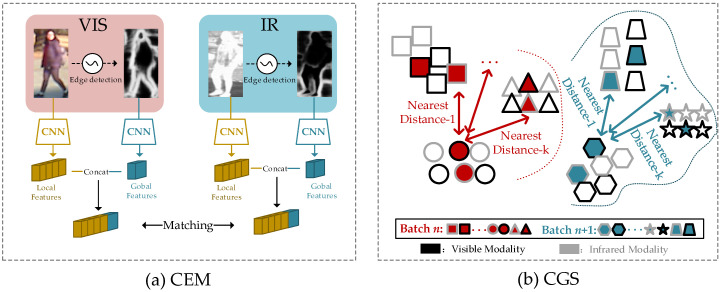
(**a**) Visible images and infrared images utilize the extended contour information obtained through contour detection to alleviate the modality discrepancy. Consequently, it becomes easier to match the same person between the visible and infrared modalities. (**b**) Different shapes represent different classes in the dataset. The CGS sampler first selects one class as an anchor. Next, it identifies the top-*k* nearest neighboring classes based on their distances to the anchor class. These selected neighboring classes are then included in the same batch for training.

**Figure 2 sensors-23-07948-f002:**
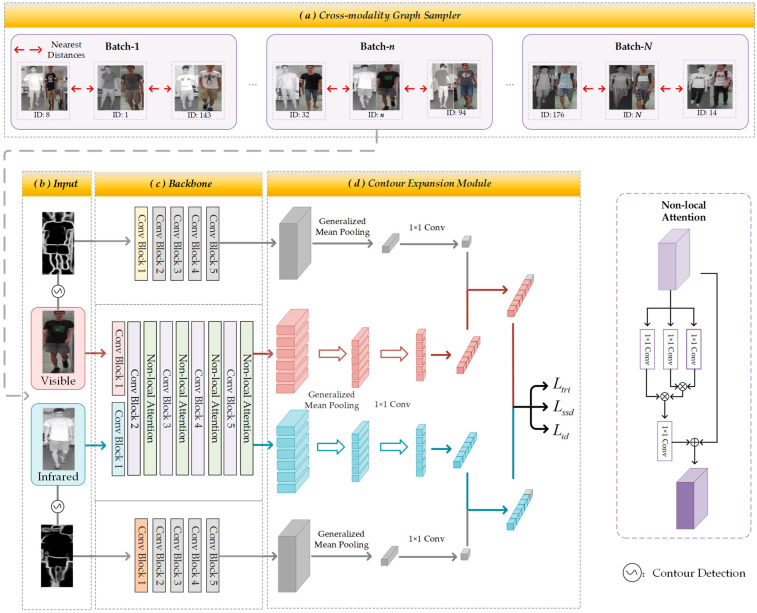
Framework of GSMEN. (**a**) CGS samples the samples into N batches separately. (**b**) The inputs are divided into three categories: visible images, infrared images, and contour images obtained through contour detection from the two modalities. (**c**) ResNet-50 [27] is introduced as the base backbone network, supplemented with Non-local Attention [28] to enhance feature extraction. (**d**) The CEM integrates local features from both Visible and Infrared modalities with global feature from the contour modality.

**Figure 3 sensors-23-07948-f003:**
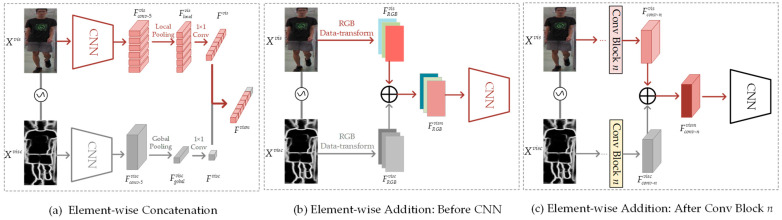
Fusion methods for contour information.

**Figure 4 sensors-23-07948-f004:**
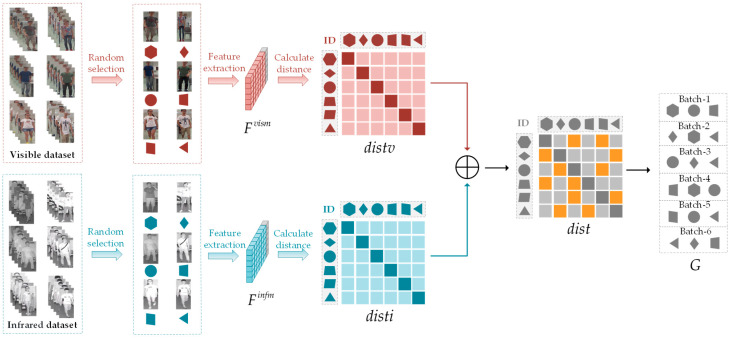
Framework of the CGS. The feature distances (distv and disti) for each class of samples in the visible and infrared modalities are obtained based on the latest trained model. Then, they are merged to obtain the cross-modal distances, denoted as dist. Next, the nearest neighbor classes are grouped into the same batch based on the distance dist to complete the sampling process.

**Figure 5 sensors-23-07948-f005:**
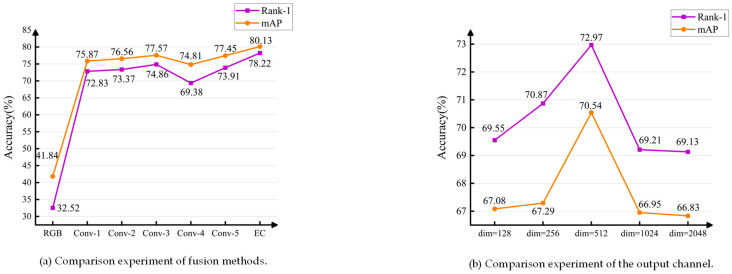
Training and testing on SYSU-MM01 dataset with All-Search mode.

**Figure 6 sensors-23-07948-f006:**
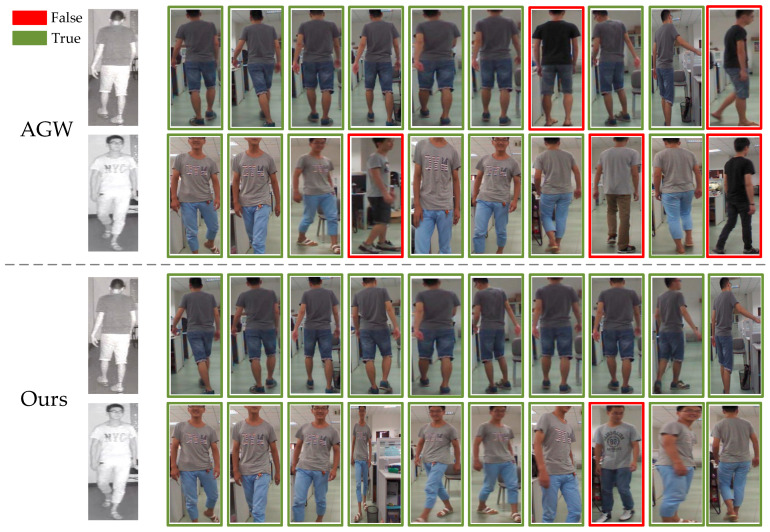
Rank-10 results of AGW and our methods. Training on SYSU-MM01 dataset.

**Table 1 sensors-23-07948-t001:** Comparing data (%) between our method and other VI Re-ID methods. Red and bold signify the best result, while blue indicates the second-best result.

Method	Venue	SYSU-MM01								RegDB							
		All-Search				Indoor-Search				VIS to IR				IR to VIS			
		R-1	R-10	R-20	mAP	R-1	R-10	R-20	mAP	R-1	R-10	R-20	mAP	R-1	R-10	R-20	mAP
cmGAN [13]	IJCAI 18	26.97	67.51	80.56	27.80	31.63	77.23	89.18	42.19	-	-	-	-	-	-	-	-
AlignGAN [49]	ICCV 19	42.40	85.00	93.70	40.70	45.90	87.60	94.40	54.30	57.90	-	-	53.6	56.30	-	-	53.40
Xmodal [17]	AAAI 20	49.92	89.79	95.96	50.73	-	-	-	-	-	-	-	-	62.21	-	-	60.18
DDAG [21]	ECCV 20	54.75	90.39	95.81	53.02	61.02	94.16	98.41	67.98	69.34	86.19	91.49	63.46	68.06	85.15	90.31	61.80
AGW [4]	TPAMI 21	47.50	84.39	92.14	47.65	54.17	91.14	95.98	62.97	-	-	-	-	-	-	-	-
cm-SSFT [45]	CVPR 20	61.60	89.20	93.90	63.20	70.50	94.90	97.70	72.60	72.30	-	-	72.90	71.00	-	-	71.70
GLMC [22]	TNNLS 21	64.37	93.90	97.53	63.43	67.35	98.10	** 99.77 **	74.02	91.84	97.86	98.98	81.42	91.12	97.86	98.69	81.06
MPANet [12]	CVPR 21	70.58	96.21	98.80	68.24	76.64	** 98.21 **	99.57	** 80.95 **	82.80	-	-	87.70	83.70	-	-	80.90
LBA [46]	ICCV 21	55.41	-	-	54.14	58.46	-	-	66.33	74.17	-	-	67.64	72.43	-	-	65.46
SFANet [50]	CVPR 21	65.74	92.98	97.05	60.83	71.60	96.60	99.45	80.05	76.31	91.02	94.27	68.00	70.15	85.24	89.27	63.77
CM-NAS [15]	CVPR 21	61.99	92.87	97.25	60.02	67.01	97.02	99.32	72.95	84.54	95.18	97.85	80.32	82.57	94.51	97.37	78.31
MMN [47]	ACM 21	70.60	96.20	99.00	66.90	76.20	97.20	99.30	79.60	91.60	97.70	98.90	84.10	87.50	96.00	98.10	80.50
MID [48]	AAAI 22	60.27	92.90	-	59.40	64.86	96.12	-	70.12	87.45	95.73	-	84.85	84.29	93.44	-	81.41
AGMNet [51]	TNNLS 23	69.63	96.27	98.82	66.11	74.68	97.51	99.14	78.30	88.40	95.10	96.94	81.45	85.34	94.56	97.48	81.19
PMT [25]	AAAI 23	67.53	95.36	98.64	64.98	71.66	96.73	99.25	76.52	84.83	-	-	76.55	84.16	-	-	75.13
GSMEN	Ours	** 72.97 **	** 98.93 **	** 99.43 **	** 70.54 **	** 78.22 **	97.38	99.24	80.13	** 93.69 **	** 98.04 **	** 99.13 **	** 92.56 **	** 91.41 **	** 98.09 **	** 98.77 **	** 92.08 **

**Table 2 sensors-23-07948-t002:** Ablation experiment results of our method. Training on SYSU-MM01 dataset. The bold indicates the best result.

Settings	All-Search		Indoor-Search	
	R-1	mAP	R-1	mAP
B	62.73	60.81	66.32	67.01
B + CEM	71.24	68.57	77.08	78.26
B + CGS	65.18	62.42	68.54	69.94
B + CGS + CEM	**72.97**	**70.54**	**78.22**	**80.13**

**Table 3 sensors-23-07948-t003:** Comparison experiments of different sampling methods. Training on SYSU-MM01 dataset. The bold indicates the best result.

Settings	All-Search		Indoor-Search	
	R-1	mAP	R-1	mAP
Random Sampler	72.02	68.73	77.18	78.54
Uniform Sampler	71.28	68.13	76.62	78.77
Cross-modality Graph Sampler	**72.97**	**70.54**	**78.22**	**80.13**

**Table 4 sensors-23-07948-t004:** Comparison experiments of different contour detectors. Training on SYSU-MM01 dataset. The bold indicates the best result.

Settings	All-Search		Indoor-Search	
	R-1	mAP	R-1	mAP
Canny edge detection [53]	71.23	67.59	70.29	74.48
SCHP [29]	**72.97**	**70.54**	**78.22**	**80.13**

## Data Availability

All datasets used for training and evaluating the performance of our proposed approach are publicly available and can be accessed from [8,39].

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
