# Peer review of "Graph Sampling-Based Multi-Stream Enhancement Network for Visible-Infrared Person Re-Identification"

_sensors, 2023, doi:10.3390/s23187948_

Round 1
Reviewer 1 Report
This paper proposes graph sampling-based multi-stream enhancement network for visible-infrared person re-identification. Besides, an innovative Cross-modality Graph Sampler (CGS) is designed for sample selection before training. Experiments are conducted to verify the effectiveness of the proposed method. This study is interesting, but some issues need be addressed before publication.
1. What is the difference between the proposed Cross-modality Graph Sampler and k-Nearest Neighbor algorithm? The proposed method is actually a classification algorithm, similar to k-Nearest Neighbor.
2. In Section 3.2, why Euclidean distances are selected for distances or similarities calculation? Hash algorithm is more commonly used for image similarity measurement.
3. Some details of the proposed method should be illustrated, such as the most basic calculation formula, rather than just providing a block diagram.
4. Lastly, I understand there are two first authors for this paper as they contributed equally to this work. But why are there two corresponding authors? Funding acquisition and Project administration were both completed by Wenfeng Zhang, why is Ruisheng Ran also the corresponding author?
1. Language needs polishing. For example, the last sentence of Abstract, what are achieved 93.69% and 92.56%? In Section 3.2, the initial letter of a sentence ’both’ should be capitalized. Other issues will not be listed one by one.
Author Response
Please note that the detailed responses are provided in the attachment for your reference. If you require a more in-depth understanding of our responses to the reviewer's comments and the corresponding revisions, kindly open the attachment for further details. Thank you for your time and attention.

Reviewer 2 Report
This manuscript proposed concatenating the contour for person re-identification and during training, the authors group classes with a similar feature to the same batch to boost the performance. Although the idea itself is straightforward, I think the manuscript is well-written and presented and the results are promising. Some minor comments are below.
1. I would suggest adding the citations in Table 1.
2. On Page-6, line-231, "both" -> "Both".
3. I think the experiments show that the contour information is beneficial, but the selection of the contour extraction method is questionable. It would be good to have an additional experiment using one classical edge detection algorithm like Canny edge detection and compare its performance to the proposed approach.
I think the language is fine.
Author Response

(The authors gave the same response as above.)
